# Analysis of Biologics Molecular Descriptors towards Predictive Modelling for Protein Drug Development Using Time-Gated Raman Spectroscopy

**DOI:** 10.3390/pharmaceutics14081639

**Published:** 2022-08-05

**Authors:** Jaakko Itkonen, Leo Ghemtio, Daniela Pellegrino, Pia J. Jokela (née Heinonen), Henri Xhaard, Marco G. Casteleijn

**Affiliations:** 1Drug Research Program, Division of Pharmaceutical Biosciences, Faculty of Pharmacy, University of Helsinki, 00100 Helsinki, Finland; 2Orion Pharma, 02101 Espoo, Finland; 3Drug Research Program, Division of Pharmaceutical Chemistry and Technology, Faculty of Pharmacy, University of Helsinki, 00100 Helsinki, Finland; 4VTT Technical Research Centre Finland, 02150 Espoo, Finland

**Keywords:** pharmaceutical proteins, Raman spectroscopy, PCA, K-means clustering, in-line measurement, biologics, protein unfolding, CD, DLS, tryptophan fluorescence

## Abstract

Pharmaceutical proteins, compared to small molecular weight drugs, are relatively fragile molecules, thus necessitating monitoring protein unfolding and aggregation during production and post-marketing. Currently, many analytical techniques take offline measurements, which cannot directly assess protein folding during production and unfolding during processing and storage. In addition, several orthogonal techniques are needed during production and market surveillance. In this study, we introduce the use of time-gated Raman spectroscopy to identify molecular descriptors of protein unfolding. Raman spectroscopy can measure the unfolding of proteins in-line and in real-time without labels. Using K-means clustering and PCA analysis, we could correlate local unfolding events with traditional analytical methods. This is the first step toward predictive modeling of unfolding events of proteins during production and storage.

## 1. Introduction

Biotechnological drugs and their development started in the early 1980s and they play an increasingly important role in the treatment of many diseases, such as anemia, cystic fibrosis, cancer, and neurological diseases [1,2,3]. Protein drugs are produced by the use of recombinant DNA technologies in expression hosts and may have post-translational modifications [4,5]. Protein-based drugs have different characteristics compared to non-biologics; they have a higher specificity resulting in greater efficacy and reduced adverse effects [6]. On the other hand, proteins are relatively fragile molecules and prone to unfolding and forming aggregates [7,8], which are a major problem in terms of efficacy, limited solubility, and increased viscosity but may also represent the main cause of immunological responses [9,10]. Their presence, nature, and amounts are thus often considered critical quality attributes [11].

To assess the incidence of unfolding problems, sensitive analytical techniques are necessary to monitor and quantify protein aggregate levels [12]. Most techniques, however, require offline measurements of (intermediate) product(s), such as:(i)The detection and characterization of (sub)visible particles (e.g., visual inspection, optical microscopy, light obscuration, flow imaging, fluorescence microscopy, conductivity-based particle counter, laser diffraction, dynamic light scattering (DLS), nanoparticle tracking analysis, MALLS, turbidimetry, and nephelometry);(ii)the use of separation techniques for the detection and characterization of aggregates, i.e., (denaturing/reducing) size exclusion chromatography, SDS/Native PAGE, capillary-SDS electrophoresis, and AF4 [13];(iii)other techniques, e.g., electron/atomic force microscopy, mass spectrometry, macro-ion mobility spectrometry, and AUC [14]. Label-free methods for evaluating protein folding states, such as infrared spectroscopy, Raman spectroscopy, UV/VIS absorption spectroscopy, fluorescence spectroscopy, and circular dichroism spectroscopy, are relevant to mention as they are utilized as in-line analytical techniques due to their non-invasive nature. A recent review outlines the importance of Raman spectroscopy to biopharmaceuticals in greater detail [15]. However, the sensitivity, robustness, and the ability of these label-free techniques for quantification need to be improved for protein applications.

One of the first reports to evaluate protein production within living cells was using fluorescent dyes combined with fluorescence spectroscopy to detect antibody aggregates in CHO cell lysates [16]. We recently evaluated the production of CNTF in living *E. coli* cells without dyes or tags using time-gated Raman spectroscopy [17]. However, the resulting spectra are complex and multivariate. Consequently, the raw data produced can be difficult to interpret.

In recent years, multivariate data analysis and preprocessing methods have considerably increased the ability to identify relevant information contained in Raman-, and other electromagnetic-spectra, for a better qualitative and quantitative analysis of biological samples [18,19,20]. Principal component analysis (PCA) and data K-means clustering are well-established techniques and allow the identification of the spectral features with the highest degree of variability [21,22,23]. K-means clustering allows the spectra to be grouped based on spectral similarity and, therefore, identify similar features and distributions. The clustering uses information contained in the individual spectra, and the results are reported as dendrograms to show the classes available. PCA is a powerful approach widely used to discriminate different Raman spectra using scores plots and enables to derive information regarding the basis of the spectral variability. These methods are suitable for handling large multidimensional data sets and exploring the complete spectral information.

Protein aggregation can, in part, be induced due to changes in secondary protein structures. Here, we evaluate protein unfolding events in vitro in a buffered, controlled environment with fluorescence spectroscopy, circular dichroism spectroscopy, and time-gated Raman spectroscopy to identify molecular descriptors of protein unfolding within time-gated Raman spectra. This first step in vitro is needed before future evaluation of protein production using Raman spectroscopy within cells has meaning. Protein aggregation was determined with DLS. The proteins taken into consideration were divided based on the characteristics of their secondary structure: α-helix, β-sheet, and α/β-mix.

## 2. Materials and Methods

### 2.1. Chemical, Reagents, and Protein Samples

Calcium chloride, 2-(N-morpholino)-ethanesulfonic acid (MES), sodium chloride, glucose, ethylene diaminetetraacetic acid (EDTA), sodium azide, silver nanoparticles—40 nm particle size (AgNP; #730807) and potassium dihydrogen phosphate were obtained from Merck Sigma-Aldrich (Darmstadt, Germany). Potassium chloride was obtained from Honeywell Riedel de Haën (Seelz, Germany) and disodium hydrogen phosphate was obtained from Fisher Scientific (Hampton, VA, USA). Buffer A (20 mM phosphate buffer saline (pH 7.6), buffer B (20 mM MES 150 mM NaCl (pH 7.4), and the silver nanoparticle solution were prepared as described before [17].

In this study, several proteins were evaluated (Table 1). BSA (Bovine Serum Albumin (7% solution; SRM 927e)) was obtained from NIST (Gaithersburg, MD, USA); F_ab_, F_(ab′)2_, IgG glycosylated (Immunoglobulin G), pepsin, ovalbumin (OVA), and *Sc*TIM (triosephosphate isomerase) were obtained from Merck Sigma-Aldrich (Darmstadt, Germany); IgG non-glycosylated was obtained from AntibodyGenie (Dublin, IRL). CNTF (ciliary neurotrophic factor) was prepared as earlier described [24,25] using buffer B; *Lm*TIM_E65Q_ was prepared as described earlier [26] and diluted with buffer A.

All protein powders, except the CNTF and *Lm*TIM_E65Q_ samples, were dissolved and diluted in buffer A prior to analysis with CD spectroscopy, DLS, tryptophan fluorescence, and time-gated Raman spectroscopy. Molecular descriptors derived from each technique are listed in Table 2.

### 2.2. Dynamic Light Scattering

The dynamic light scattering (DLS) was performed using a Zetasizer APS (Malvern Pananalytic, UK) with a 96-well plate autosampler at a 90° fixed angle and using an internal heat-controlled measuring cuvette. All protein samples (60 µL; final concentration 0.2 mg/mL) were filtered with a 0.45 µm syringe filter and diluted in buffer A and kept on ice prior to measurements. The pepsin solution was milky prior to dilution. Thus, a 1 mg/mL pepsin solution was spin-filtered prior to dilution with a 0.22 µm filter (Merck-Millipore) at 12,000× *g* for 4 min to remove additional impurities. To assess the aggregation behavior of protein samples upon heating, Rh measurements with DLS were carried out during thermal ramping (2–80 °C). Samples were heated from 2 to 80 °C with a 1 °C step size to determine the point of aggregation. The measured data were collected and analyzed with the Zetasizer Software (NANO, μV, APS) v6.02 (Malvern Panalytical; Malvern, UK). Three parameters were evaluated: the Z-average of the hydrodynamic diameter (automatic evaluation; *n* = 3), the lowest value of the hydrodynamic diameter (manual evaluation; *n* = 1), and the polydispersity index (PDI).

### 2.3. Tryptophan Fluorescence

Protein samples were diluted in buffer A to 0.2 mg/mL and each sample (100 µL in triplicate) was heated for 10 min in a water bath (VWR, PA, USA) at 25, 30, 37, 40, 45, 50, 52, 55, 57, 60, 65, 67, 70, 75, and 85 °C and placed on ice prior to measurement. The *Lm*TIM_E65Q_ was subjected to higher temperatures due to its known higher thermostability (25, 50, 55, 60, 65, 70, 75, 78, 80, 83, 86, 90, 95, and 99 °C. The samples were analyzed in white closed-bottom microtiter plates (Hamilton, Reno (NV), USA) with a Thermo Scientific Varioskan LUX (ThermoFisher Scientific, Waltham (MA), USA). IgG, BSA, OVA, and F_ab_ samples were excited at 295 nm and the emission spectrum were recorded between 314–550 nm, while *Lm*TIM_E65Q_ was excited at 280 nm and the emission was recorded at 290–550 nm.

### 2.4. Circular Dichroism (CD) Spectroscopy

Samples were diluted to 0.1 mg/mL with deionized, sterile water from 1 mg/mL solutions. A Chirascan CD spectrometer (Applied Photophysics, Leatherhead, UK) was used to collect CD-spectroscopy data between 22 °C and 90 at 280 nm using a 0.1 cm path-length quartz cuvette. Data were collected every 1 nm utilizing 1 s as the integration time. Each measurement was performed in triplicate with baseline correction. Pro-Data Viewer software SX v2.5.0 (Applied Photophysics, Leatherhead, UK) was used to analyze the spectra. The melting temperature was determined via thermal unfolding of the protein sample between 190 and 260 nm with a 2 °C step size at 1 °C/min ramp rate with ±0.2 °C tolerance and subsequently analyzed with the Global3 software package v3.1(Applied Photophysics, Leatherhead, UK).

### 2.5. Time-Gated Raman Spectroscopy

The time-resolved measurements were performed as described earlier [17,28,29], with minor alterations. The time-gated Raman measurements (time-gated) were performed with a reduced laser power of approximately 20 mW (checked with an Ophir Nova II laser power meter, Ophir Optronics Solutions Inc., Jerusalem, Israel) at the sample to avoid photo-bleaching. Data acquisition software and setup control were carried out by Timegated instruments software (Timegate Instruments Oy, Oulu, Finland). Protein samples were prepared as described in Section 2.3, except for the IgG non-glycosylated sample. All protein measurements, except for IgG non-glycosylated, were performed at ambient temperature and humidity, and spectra were the sum of 11 repeats at an acquisition time of 14 min. Two separated detector ranges (1900–900 and 1400–400 cm^−1^) were measured in such a way. Protein samples (50 µL) were measured in a custom-made aluminum round-bottom well [17], positioned on top of a 3D-printed plastic holder within the Timegated Instruments Sample-Cube using the Timegated a common BWTek sampling probe.

The analysis of the IgG non-glycosylated sample was performed using a 3D-printed heat-ramping prototype (Appendix A) attached to a Julabo’s SL-26 heat circulating water bath. Between the 3D-printed plastic, its aluminum cover, and the time-gated sampling area, 420 µL of distilled water was covered with a 0.13-0.16 mm thick cover glass (Paul Marienfeld GmbH & Co., Lauda-Königshofen, Germany), sealed with pivot grease (Eppendorf, Hamburg, Germany) to ensure optimal heat transfer. The sample (40 µL) was applied in an aluminum crucible (40 µL ME-51119870; Mettler Toledo, Switzerland) and sealed with 0.13–0.16 mm thick cover glass (Paul Marienfeld GmbH & Co., K, Lauda-Königshofen, Germany) and pivot grease (Eppendorf, Hamburg, Germany) and then placed on the Time-gated sampling area. Heat ramping was performed according to Appendix A. The time-gated spectra were the sum of 6 repeats (1811–555 cm^−1^) at an acquisition time of 6.05 min per temperature. The temperature in the crucible in the time-gated sampling area was checked with a FLIR TG165 imagining infrared thermometer prior to Time-gated sampling to ensure the correct temperature at the Time-gated sampling site. The thermometer had an error of ±1.5%, or a minimum of 1.5 °C, and the emittance was set to 0.25 according to the manufacturer’s recommendations and was in line with a roughened aluminum surface [30].

### 2.6. Data Preprocessing

The implementation of multivariate data analysis methods requires pretreatment of the raw data. The preprocessing helps to eliminate unwanted signals and to enhance the discrimination of structural features [20].

All spectral data (Table 2) was collected as text comma-separated values files to facilitate the manipulation and provide a graphical representation of the data. Data analysis was performed using R (R-Studio, Boston (MA), USA). Before statistical analysis, the data sets have been corrected for baseline and vector normalized to facilitate comparison.

### 2.7. Data Analysis

K-means clustering analysis is an unsupervised learning algorithm widely used for spectral image analysis [18]. In summary, it partitions the observations into clusters, with the cluster centroid representing the whole cluster. The pre-processed spectra are grouped according to their spectral similarity, forming clusters for particular temperature points, each characterizing regions of the image with similar molecular properties. The dissemination of similarity can be visualized over the sample image or as a dendrogram showing the hierarchical relationship between classes. Along with the spectra, additional parameters like the number of clusters (k) and the initial cluster centers are calculated. The centroids are set by shuffling the dataset and randomly selecting K points, and then each point of the dataset is associated with the nearest centroid. The process is repeated until there is no change in the centroids. Eventually, k clusters with the most similar spectra present themselves and the centroids are determined by taking the average of all data points that belong to each group.

PCA is a multivariate analysis broadly used to reduce the dimensionality of large data sets [18]. It is mainly used to represent a multivariate data table as a smaller set of variables to identify trends, jumps, clusters, and outliers. PCA identifies a new coordinate system in the K-dimensional space that maximizes variation in the data space. This reduction help discover relationships between observations and variables and among the variables. Important information is extracted from the data and expressed as a set of indices called principal components. The importance of each PC is identified by ordering the Eigenvalues in descending order, corresponding to the descending order of variance, and denotes their importance to the dataset. The PCs contribute less in decreasing order; the first PCs contain the most information. The loading of a PC provides information on the source of the variability inside the spectra, derived from variations in the molecular components recorded from different experiments. The pre-processed dataset was evaluated by PCA, with the covariance matrix to represent the dataset by eigenvectors accounting for most of the variance and identifying the spectra’ similarities. Usually, the first two or three PCs represent the highest variance present in the data sets [19].

K-means clustering and PCA were performed on the preprocessed time-gated Raman spectral data of each protein (graphs are shown in the Appendix A). The dendrograms (panel 1 in the Appendix A) show the different classes available, grouped by temperature. The cluster means (panels 2 in the Appendix A) depict the variance and the contribution of the first components. The Scree plots (panels 3 in the Appendix A) corresponding to each PC obtained by PCA have peaks that can be attributed to the protein constituents and show the region of the Raman spectra where the main differences occur (panels 4 in the Appendix A). Their respective negative and positive loadings contribute substantially to the differentiation of the protein structure. This enables one to derive information regarding the basis of the spectral variability. PCA provides thus insight into the source of the spectral variability and, therefore, the differentiation of the protein’s structural components.

## 3. Results

### 3.1. Alpha Helical Proteins

CNTF is a small α-helical, dimeric protein of 22.8 kDa and comprises 4 α-helical bundles per monomer [24]. The aggregation and unfolding of CNTF, studied by DLS and CD (Appendix A), is described in a recent publication [27], and starts at 38 °C, with an estimated T_m_ of 53 °C (Table 3, Table 4 and Table 5). Our CD data is in line with this earlier observation and displays a typical band at 220 nm (Appendix A), which is due to the peptide n → π* transitions and is indicative of α-helical structures [31]. In the tryptophan fluorescence spectra, upon unfolding, CNTF exhibits a redshift of approximately 10 nm, indicating the movement of the polar groups from a hydrophobic environment to a hydrophilic environment. In addition, the fluorescence spectra showed a lowering trend in the intensities as the temperature increased.

BSA comprises 3 α-helical domains with a molecular weight of 66.5 kDa [32]. The α-helical structure is evident from the CD spectra, where the negative band at 208 nm, present due to the exciton splitting of the lowest peptide π → π* transitions, is more prominent due to 3_10_ α- helix structures (Appendix A) [31]. BSA aggregation, as evaluated with DLS, starts at ~58 °C, where we observed an initial increase in intensity (data not shown) and size (Appendix A). After a short plateau, the particle size of the aggregates increased swiftly, which is even observable in the sample polydispersity (Appendix A).

From the tryptophan fluorescence spectra (Appendix A), we observed a redshift of approximately 10 nm, indicating the movement of the polar groups from a hydrophobic environment to a hydrophilic environment, even though the intensities of the BSA spectra were relatively low at 25–30 °C and with intensities highest at 45–50 °C, to then come down to the same level of intensity slowly as 25 °C at temperatures above 70 °C.

The time-gated Raman spectra of BSA showed a clear drop in intensity between 55 and 57 °C (Appendix A), especially around 1650 cm^−1^. In addition, the spectra grouped according to their similarity shown in Appendix A panel 1, showed that the spectra recorded at 25–55 °C differed from the spectra recorded at 57–85 °C. This difference correlates with the start of aggregation seen in DLS. The two groups were well discriminated in the cluster means, as shown in panel 2 (Appendix A). PC1 and PC2 contributed to most of the explained variance and allowed the discrimination between the two groups. Raman peaks were identified that contribute to the PC scores. The positive and negative correlation of PC1 and PC2 is depicted in Appendix A (panel 4), where zero is the dashed line. At Raman peaks, where the spectral differences between data exist (i.e., the correlation of PC1 and PC2 is in the opposite direction), the corresponding physical changes in protein bonds were relevant changes due to thermal unfolding. The significant differences in the secondary structure of BSA due to the increase in temperature are summarized in Appendix A.

### 3.2. Beta Sheet Proteins

One example of a β-sheet protein is pepsin A, an archetypal aspartic proteinase belonging to the class of endopeptidases. The aspartic proteinases display a predominant β-fold with only a few short helical segments [33], though it has been classified by Rygula et al. (2013) [34] as β-sheet protein. This was evident from the CD-spectra (Appendix A), as the spectrum was a mix of mainly random coil and β-sheets. At the same time, the protein appeared to be disordered at pH 7.0. Pepsin started aggregating at this pH at about 64 °C (Appendix A). The main observation from the tryptophan fluorescence spectra at different temperatures (Appendix A) was that the intensities were very low for all spectra, and as such, no conclusions can be drawn.

The time-gated Raman spectra of pepsin showed no clear drop in intensity due to the increase in temperature (Appendix A). As such, the spectra grouped according to their similarity at different temperature points shown in Appendix A (panel 1) did not show significant differences. This corresponded to the observations in the CD-spectra that the protein, in this particular low-quality sample, was already partly unfolded at room temperature. PC1 and PC2 contributed to only 50% of the explained variance. Raman peaks were identified that contributed to the PC scores. However, the large changes in the relative intensities observed in the grouping in Appendix A panel 1 show that the spectra at clusters 25, 30, 40, 45, 75, and 85 °C and at clusters 37 and 50–70 °C show similarities. These observations did not correspond with the aggregation temperature of 64 °C, indicating further that the pepsin protein solution used in this study was of poor quality.

The positive and negative correlation of PC1 and PC2 is depicted in Appendix A (panel 4), where zero is the dashed line. At Raman peaks where the spectral differences between data exist (i.e., the correlation of PC1 and PC2 is in the opposite direction), the corresponding physical changes in protein bonds are relevant changes due to thermal unfolding. Changes in the secondary structure following the increase in temperature can be observed in Figure 1, Appendix A, and Appendix A, and summarized in Appendix A.

When directly comparing the relative time-gated Raman spectra of pepsin at ambient temperature and 65 °C (Figure 1), we observed changes in the secondary structure upon aggregation. It is also evident that at ambient temperature, pepsin was already partly unfolded, indicated by the peak at 1245 cm^−1^, typical of random coil structures. The fermi doublet ratio of ~1 showed that the buried tryptophans were more exposed to the hydrophilic matrix. The ratio of the tyrosine peaks at 850/830 cm^−1^ = 0.65 at 25 °C to 0.43 at 65 °C showing tyrosines act as a strong H-bond donor with little change between these temperatures.

Another class of β-sheet proteins are antibodies or their fragments [35,36], which during the last decades, have proven themselves as a highly effective and specific class of biological drugs if they maintain a high thermostability and low aggregation propensity [37]. Their structures are very well characterized and proteolytic cleavage to remove the Fc tail results in either F_ab_ or F_(ab′)2_ fragments [35]. In addition, IgG antibodies are naturally modified by the decoration of glycan sugars [38]; however, deglycosylated IgG has its place as a therapeutic as well [39]. The β-sheet secondary structure was well characterized by the CD spectra of both the F_ab_ fragment and the full IgGs (Appendix A). We derived the clear right twisted anti-parallel β-sheet formation [40] as expected in the F_ab_ fragment and the non-glycosylated IgG spectra [41] as they consist of seven β-strands with four strands forming one β-sheet and three strands forming a second sheet. However, glycosylated IgG appeared to be more relaxed or exhibited left twisted anti-parallel β-sheets (Appendix A). The stability of IgG, or its derived fragments, appeared similar based on their melting temperatures (Table 3); however, individual F_ab_ fragments appeared to aggregate at slightly lower temperatures (Appendix A and Table 5). Tryptophan fluorescence spectra comparing the F_ab_ fragments and the full IgG appeared very similar (Appendix A).

However, when overlaying the time-gated Raman data of glycosylated and non-glycosylated IgGs, there were clear differences at 20 °C (Figure 2A). Since aggregation had its onset at ~64 °C (Table 4), we also compared time-gated Raman spectra at 65 °C (Figure 2B), where the changes differed in response to the rise in temperature. We observed a clear shift in the amide I peak from 1631 to 1646 cm^−1^ due to the loss of β-sheet structures [42] and a reduction of the 1097 cm^−1^ peak. Yet, at ambient temperatures, both spectra differed greatly. In non-glycosylated IgG time-gated spectra, we observed changes at 954/986 cm^−1^ (likely the protein backbone) and the 1207 cm^−1^ peak (cysteine or the νSO_4_ peak) [43]. To our surprise, we observed the very characteristic carotenoid peaks at 1152 and 1517 cm^–1^ due to C–C and conjugated C=C bond stretches [44], on which we can only speculate them to be a remnant of the production process.

In the glycosylated IgG spectra (Figure 2A,B), we observed the reduction of the C-H (def) peak at 1456 cm^−1^, Trp Cα-H (def) peak at 1455 cm^−1^, and changes in 1045 cm^−1^, 1097 cm^−1^, all likely marker bands of aromatic side chains affected upon heating. The relative Raman intensities between 600–900 cm^−1^ appeared to drop dramatically; however, this could also be an artifact due to the detector shift between lower and higher wavenumbers (Raman shift) in this particular measurement. Despite these differences, the peak at 606 cm^−1^, likely the CCC deformation in-plane vibration mode of the phenylalanine ring, in the non-glycosylated spectra did not change due to heating [45]. Of interest is the ratio of the Raman peak intensity seen in the tyrosine doublet Raman bands near 850 and 830 cm^−1^ of 0.89 at 20 °C and 1.14 at 65 °C. This shift indicates that upon heating, the 10–12 tyrosines present in the IgG were more exposed to the solvent, albeit the shift is small.

In the non-glycosylated IgG sample, we observed the disappearance of the carotenoid peaks, the likely protein back-bone peaks at 954/986 cm^−1^, the 1207 cm^−1^ peak, and the C–N peak at 1120 cm^−1^. Overall, we saw fewer changes in the overall spectra compared to glycosylated IgG. The Int_850_/Int_830_ ratio of tyrosine shift from 0.89 at 20 °C to 1.53 at 65 °C indicates a higher exposure of tyrosines to the solvent upon heating.

**Figure 2 pharmaceutics-14-01639-f002:**
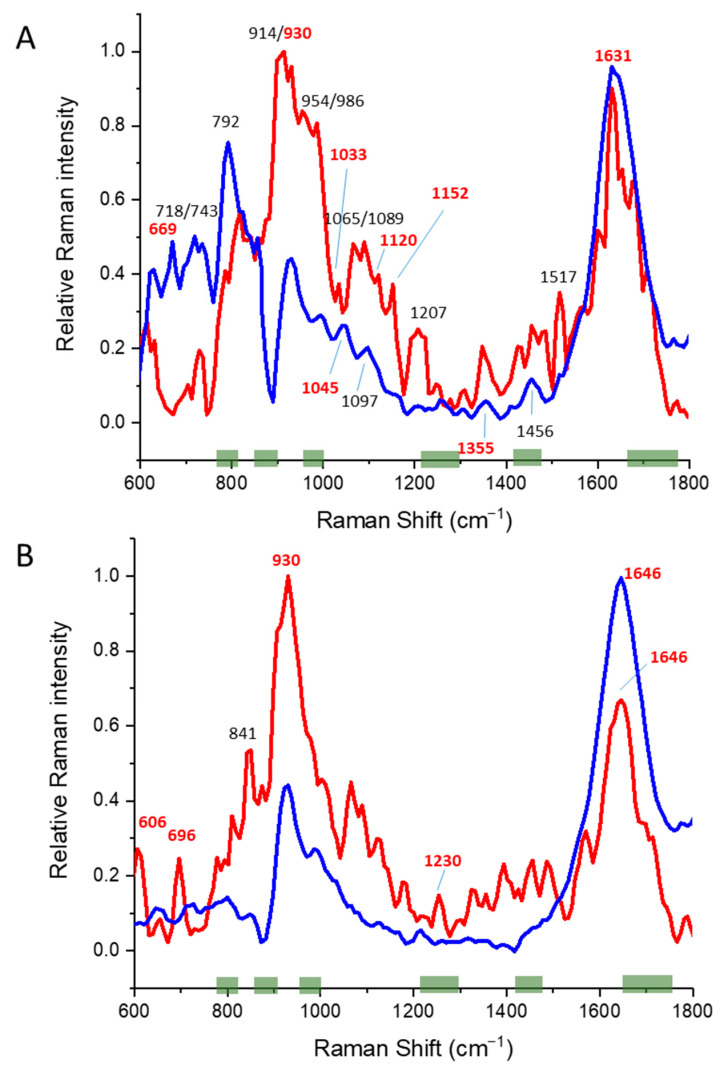
Averaged and normalized Time-gated spectra of IgG (glycosylated; blue; N = 11) and IgG (non-glycosylated; red; N = 6) at (**A**) 25 °C, (**B**) and at 65 °C. Individual, non-processed, and non-normalized Time-gated spectra (i.e., raw data) are shown in Appendix A. Regions of significance in respect of the glycosylation status of proteins, as determined by Brewser et al. (2011) [46] are shown in green. Raman values of significance, as presented in Appendix A, are depicted in bold and red.

The time-gated Raman spectra of glycosylated IgG did not show a clear drop in intensity in the amide I peak (1650–1660 cm^−1^) due to the increase in temperature (Appendix A). The intensity rose first and then dropped at the higher temperatures. As with pepsin, the spectra grouped according to their similarity at different temperature points shown in Appendix A (panel 1) do not show any differences. However, there is no indication in the CD-spectra that the protein was partly unfolded (Appendix A). The one high intensity at 50 °C at 880 cm^−1^ indicated a specific large change in the tryptophan environment. In addition, the grouping in Appendix A panel 1 shows that the spectra at clusters 25, 30, 40, 37, 40, 45, 60, 70, and 85 °C and at clusters 50, 52, 55, 57, 65, and 75 °C show similarities. Yet, PC1 and PC2 contributed around 50% of the explained variance, so it does not allow the discrimination between the two groups. These observations did not directly correlate with the aggregation temperature of 64 °C and CD melting temperatures (T_m_) of 65.5 and 72.1 °C. Upon closer inspection in Appendix A panel 2, we observed the second cluster was grouped due to changes around a Raman shift of 1460–1480 cm^−1^ (C–H and aliphatic side chains) at these temperatures. PC1 and PC2 contribute around 60% of the explained variance. Significant changes in the IgG secondary structure due to the increase in temperature can be observed in Appendix A, summarized in Appendix A.

Summarizing, for β-sheet proteins, an interesting observation is the relevance change of C=O stretching peak between 1760–1840 cm^−1^, most likely in the carbonyl groups upon thermal unfolding. Overall, the intensity decrease in the amide I peak is likely due to the increased interaction of amino acids with water, thus indicating the unfolding of the β-sheet proteins. In addition, significant changes observed both in IgG type structures and pepsin are changes due to tryptophan in the fingerprint region. The latter observation correlates with the major changes observed in the tryptophan fluorescence spectroscopy.

### 3.3. Alpha/Beta Proteins

Ovalbumin is the main protein found in egg white (~55% of the total protein) and consists of 385 amino acids, with a relative molecular mass of 42.7 kDa. Ovalbumin contains several post-translational modifications, including N-terminal acetylation (G1), phosphorylation (S68, S344), and glycosylation (N292). Ovalbumin’s internal signal sequence (residues 21–47) is not cleaved off but remains as part of the mature protein. Ovalbumin displays sequence and three-dimensional homology to the serpin superfamily, but unlike most serpins, it is not a serine protease inhibitor.

The secondary structure, also supported by the CD-spectrum (Appendix A), was comprised of α-helices (12) and β-sheets (15). The β-sheets form the core of the protein, while the α-helices form the outside of the protein, especially in its dimeric form [47].

Ovalbumin aggregation could not be evaluated with DLS in this study as the samples seemed contaminated with larger molecules or were already partly aggregated prior to heating in the DLS-cuvette (Appendix A). Earlier reports indicated aggregation to start at ~71 °C [48]. According to the tryptophan fluorescence spectra (Appendix A), the intensities of the spectra were highest between 37–40 °C and 50 °C, with a lower intensity at the highest temperatures (75–85 °C). These results indicate that neighboring amino acids initially reduced quenching, while the temperature-induced changes in the secondary structures then increased quenching. In addition, we observed a blue shift, and since W149 and W268 are buried at room temperature and bound to charged groups, water molecules may create a blue shift in this environment [49]. W185 is buried and stabilized by hydrophobic groups [47].

The time-gated Raman spectra (Appendix A) show a dramatic drop in intensity of the amide I peak between 40–50 °C, indicating denaturation [50], with some recovery at 60 °C. In addition, the spectra grouped according to their similarity at different temperatures shown in Appendix A panel 1, show that the spectra from 25–40 °C were more similar than the spectra from 55–85 °C. Combined with the tryptophan fluorescence data, it appears that our sample started denaturing at lower temperatures than earlier reported [48].

Significant changes in the secondary structure of ovalbumin following the increase in temperature were observed in the time-gated Raman spectra (Figure 3 and Appendix A) and are summarized in Appendix A. Upon heating, we observed the reduction of α- helical structures, as indicated by the amide I peak splitting and the rise of the tryptophan indole ring peak at 1561 cm^−1^ (Figure 3A). Furthermore, due to the overall reduction of the peak intensity of the amide I peak, the peak details seen in the normalized data appear enhanced in the heated sample. PCA analysis of the time-gated Raman spectra in Appendix A, panel 4, identified the relevant changes due to a positive correlation in the PC1 components of Raman peaks of phenylalanine (at 1015–1020 cm^−1^), the amide II (at 1200 cm^−1^) and C=O stretching bond (at 1750–1860 cm^−1^). Further modeling of the amide I peak during thermal unfolding could give a better insight into the unfolding of ovalbumin.

Triosephosphate isomerase (TIM) comprises the classical α/β barrel [51] and the wild-type exists as a dimeric protein. The *Leishmania mexicana* (*Lm*) mutant E65Q (*Lm*TIM_E65Q_) is a thermostable variant of the wild-type protein [26]. This mixed α-helix/β-sheet structure is clearly observed in the CD-spectrum (Appendix A), and the *Lm*TIM_E65Q_ variant appears to be better folded than Saccharomyces cerevisiae (*Sc*) TIM under the same conditions. *Sc*TIM aggregation evaluated with DLS in this study (Appendix A) indicates aggregation to start at ~58 °C, which was in correlation with the CD-melting curve (Table 3). In the tryptophan fluorescence spectra (Appendix A), the intensities of the spectra were highest between at 37–40 °C, with a lower intensity at the highest temperatures (75–85 °C). These results indicate that neighboring amino acids initially reduced quenching, while the temperature-induced changes in the secondary structures then increased quenching. In addition, we observe for both TIMs a redshift above 300 nm^−1^, indicating exposure of the tryptophans to the solvent. In *Sc*TIM, both W89 and W156 are buried and stabilized by hydrophobic groups [52], while in the *Lm*TIM_E65Q_ W11, W160, and W194 are buried and stabilized by hydrophobic groups, while the buried W91 is bound to a polar group [53]. Both the W167 in *Sc*TIM and W170 in *Lm*TIM_E65Q_ reside in the hinge of the catalytic loop and thus are exposed to the solvent [54].

The time-gated Raman spectra show a large reduction in relative intensity between 83 and 86 °C for *Lm*TIM_E65Q_ (Appendix A), which is in accordance with the CD melting curve and tryptophan fluorescence measurement (Appendix A). In addition, the spectra grouped according to their similarity shown in Appendix A panel 1 show that the spectra from 25–86 °C were more similar than the spectra from 90–99 °C. However, spectra grouped in Appendix A, differ, as we observed earlier in the IgG spectra, due to changes around Raman Shift of 1460–1480 cm^−1^ (C–H and aliphatic side chains) at 25, 30, 40, 45, 75, and 85 °C. PC1 and PC2 contribute to most of the explained variance and allows the discrimination between the two groups.

Significant changes in the secondary structure due to the increase in temperature in *Lm*TIM_E65Q_ were observed in time-gated Raman spectra (Appendix A; summarized in Appendix A). Upon heating, we observed the changes in the secondary structure of the α/β barrel, as indicated by the significance of changes in the amide II peaks at 1285 cm^−1^ (α helix/ β-sheet) and 1230–1240 cm^−1^ (β-sheet) (Figure 3A). Furthermore, due to the reduction of the peak intensity of the amide I peak, details in other regions of the data were enhanced. The unfolding of this variant of *Lm*TIM is well understood [53]. At neutral pH, the active dimer unfolds into partially unfolded monomers, which are prone to aggregation. Hence, the relatively small changes in the amide I peak, compared to ovalbumin, are due to remaining partially folded monomers. Unlike in the ovalbumin spectra, we observed significant changes in the peptide bonds in *Lm*TIM_E65Q_.

When comparing common features of unfolding in ovalbumin and TIM, we observed that the K-clustering profiles correlated well with the tryptophan fluorescence. In both cases, when the red shift occurs and the intensities drop, we also observed a reduction in the amide I peaks in the time-gate Raman spectra. The relevant changes derived from the PCA analysis observed for both proteins (Appendix A, and Table 6) are due to the phenylalanine peak 980–1020 cm^−1^, the amide II peak (1200–1205 cm^−1^), and the C=O stretching bond between 1820–1860 cm^−1^.

## 4. Discussion

Pharmaceutical proteins have proven to be very important in the field of medicines and vaccines [1]. The practice of pharmacovigilance has gained significant momentum since 1963, following the thalidomide tragedy [55]. As such, the evaluation of proteins during biotechnological production, downstream processing, and storage are crucial. Since protein function is linked directly to their three-dimensional shape or structural fold, the evaluation of changes in their secondary structure during the above-mentioned processes sheds important insights into their stability, which is directly linked to safety and efficacy [56].

Earlier, we addressed the notion that there is a need to directly monitor the intermediate products during protein production within the living cells [17]. Before we can utilize Raman spectroscopy for this task, we first need to understand how protein Raman spectra change in the function of unfolding and aggregation. In this study, we induced changes in the secondary structure of several proteins in in-vitro conditions via thermal ramping. Then we used time-gated Raman spectroscopy coupled with PCA analysis of the Raman spectra to evaluate relevant molecular descriptors toward predictive modeling of unfolding and aggregation of pharmaceutical proteins. Furthermore, we created a novel 3D-printed heat-exchange Raman sample holder for expensive samples. We evaluated several proteins to find common molecular descriptors within time-gated Raman spectra regarding the thermal unfolding of proteins and compare the changes in Raman spectra with established spectroscopic methods often used to evaluate changes in the secondary structure of proteins and the formation of aggregates. Finally, we could identify different changes due to thermal unfolding between the three protein classes (α, β, α/β). For each protein, a deeper analysis using NMR unfolding studies and additional unfolding studies using different unfolding methods would be very insightful for each specific protein; however, this was not the aim of this study.

Overall, CD measurements are in accordance with earlier reports (references are listed in Table 3), taking into account differences in concentrations or buffer. The DLS results, as summarized in Table 4, show insight into the start of aggregation and cannot be compared to CD melting curves; however, similar trends can be observed upon heating the different proteins. The tryptophan-fluorescence measurements, summarized in Table 5, showed how the local environment of tryptophan changes during unfolding and aggregation. Changes in intensities are due to quenching events [57], while red and blue shifts are due to changes in specific interactions with tryptophans within the protein [49]. We observe all these effects, but not tryptophan oxidation [58].

**Table 3 pharmaceutics-14-01639-t003:** Summarized CD results.

Protein (α, β, α/β)	T_m_ (°C)	Van’t Hoff Enthalpy (kJ/mol)	Literature Value T_m_ (°C)
BSA (α)	61.2 ± 0.1 ^a^	256.6 ± 8.4	63 [59]
CNTF (α)	55.0 ± 1.9 ^b^	360.0 ± 34.3	53 [27]
F_ab_ (β)	73.9 ± 0.3	437.5 ± 17.4	61–70 [60,61]
IgG_glycosylated_ (β) ^c^	65.5 ± 0.2 72.1 ± 0.4	336.6 ± 7.9 203.9 ± 10.6	60–68 71–77 [60,62]
IgG_non-glycosylated_ (β)	71.5 ± 0.2	265.2 ± 12.6	62–66 [63]
Pepsin (β)	49.9 ± 0.2 ^d^	352.1 ± 17.0	52 ^e^ [64]
Ovalbumin (α/β)	72.3 ± 0.1	181.3 ± 2.2	71–76 [65]
*Sc*TIM (α/β)	55.3 ± 1.7	360.0 ± 13.4	~58 [66]
*Lm*TIM_E65Q_ (α/β)	81.0 ± 0.3 ^a^	172.9 ± 12.6	83 [26]

^a^ The protein was not totally unfolded at 92 °C; ^b^ Tighter α-helical folding occurred between 30–40 °C; ^c^ Additional unfolding at 35 °C; ^d^ Protein might be aggregated to disordered at pH 7 during analysis; ^e^ At pH = 8.0.

**Table 4 pharmaceutics-14-01639-t004:** Summarized DLS results.

Protein (α, β, α/β)	Hydrodynamic Diameter at 20 °C [nm]	T_aggregation_ [°C]	Literature Value T_aggregation_ [°C] ^c^
BSA (α)	8.0	58	~62 [67]
CNTF (α)	ND	ND	38 ^a^ [27]
F_ab_ (β)	205.8	60	-
F_(ab′)2_ (β)	11.2	63	-
IgG_glycosylated_ (β)	12.3	64	55–80 [68]
IgG_non-glycosylated_ (β)	ND	ND	-
Pepsin (β)	50.9	64	-
Ovalbumin (α/β)	28.8	- ^b^	71 [48]
*Sc*TIM (α/β)	68.4	58	-
*Lm*TIM_E65Q_ (α/β)	ND	ND	-

^a^ Hydrodynamic radius at 2 °C is 2.42 ± 0.50 nm and 2.95 ± 0.22 nm (two different buffers were used [27]); ^b^ Could not be determined (see Appendix A); ^c^ To best of our knowledge.

**Table 5 pharmaceutics-14-01639-t005:** Summarized tryptophan fluorescence results.

Protein (α, β, α/β)	Maximum Fluorescence Intensity at Temperature [°C]	Red/Blue Shift ^a^	Tryptophan Oxidation (Peak at 515)
BSA (α)	45	Red (10 nm)	No
CNTF (α)	30	Red (12 nm)	No
F_ab_ (β)	85	Red (4 nm)	No
F_(ab′)2_ (β)	ND ^b^	ND	ND
IgG_glycosylated_ (β)	85	Red (5 nm)	No
IgG_non-glycosylated_ (β)	ND	ND	ND
Pepsin (β)	37	No	No
Ovalbumin (α/β)	37 50	Blue (5 nm) Blue (5 nm)	No
*Sc*TIM (α/β)	65	Red (10 nm)	No
*Lm*TIM_E65Q_ (α/β)	86	Red (6 nm)	No

^a^ Comparing the maximum of the peaks at lowest temperatures to the highest; ^b^ ND = Not Determined.

**Table 6 pharmaceutics-14-01639-t006:** Summarized results of changes in the time-gated spectra identified with K-means clustering and PCA analysis.

Protein Structural Class	Most Relevant Changes [cm^−1^]	Bond Type	Relevant Temperature Change Correlates with
α	880–900	Trp	Trp-fluorescence
	910	Ser ^a^	DLS
	940	N-Ca-C	CD
	950	N-Ca-C	
	970	Ser/His ^a^	
	1180	Val/Arg/other amino acids ^a^	
	1280	Amide II	
	1310	Phe, Tyr, Trp ^a^	
β	1350–1390	Trp ^a^	Trp-fluorescence
	1695–1760	Amide I/carbonyl stretch	
α/β	980–1020	Phe	Trp-fluorescence
	1200–1205	Amide II	DLS
	1820–1860	C=O	CD

^a^ According to De Gelder et al. (2007) [69].

Changes in the time-gated spectra are related to changes in the secondary structure of the protein. While the intensity reduction of the peptide-bond (C–N) peak would clearly indicate protein degradation and similar changes of the phenylalanine peak at ~1000 cm^−1^ would indicate chances in the protein concentration, other changes are more subtle and a reflection of secondary structure changes. Table 6 summarizes shared regions per protein structural class, though only one α-helical protein was evaluated.

One drawback of the TimeGated^TM^ device used in the study was a baseline artifact due to switching the measuring window from a lower to a higher range. We chose this set-up to maximize the number of datapoints and thus lower the deviation. We intentionally do not correct for the baseline in this study as we were aiming for a method that can be utilized by only rapidly evaluating the raw data. However, if changes in both regions were significant, the PCA analysis isolated changes as significant. The difference in intensity between the low and the higher wavenumber scans of the same sample is due to the higher intensity from the Amide I peak during the second scan. In the non-glycosylated sample, we opted for larger steps to cover the same spectrum in one scan, but here we have fewer repetitions per scan and thus introduce a larger standard deviation.

When we compare time-gated Raman data of (partly) unfolded CNTF [17] with the thermal unfolding of BSA, we observed β-sheet formation due to the rise of a peak at 1332 cm^−1^ and changes in the amide III peak of tyrosine. The most significant changes in BSA are in the backbone, in the secondary C-N peak at 1130 cm^−1^, and in the peak at 1180 cm^−1^ (Table 6). Unlike the other two protein classes, no common features can be observed other than tyrosine peak changes at 1240 cm^−1^ (amide III (β-sheets); Table 6), thus shifting away from α-helical structures.

Finally, we combined all time-gated Raman spectra in the PCA analysis to utilize the noise filtering properties of the method on the lower-scanning region of the detector of the TimeGated^TM^ device used in the study (500–1300 cm^−1^). The positive and negative correlation of PC1 compared to PC2 for all the Raman peaks combined to show three regions of particular interest (Appendix A). Changes in the peaks in, or close, to the amide III region (1180–1300 cm^−1^), tryptophan (880 cm^−1^), and phenylaniline (~1000 cm^−1^) were present in all proteins due to thermal unfolding.

Rather than dissecting each protein structure in detail to identify the structural changes during the unfolding and aggregation events as observed in the time-gated Raman spectra, we aimed to reduce the raw data to smaller components to identify significant changes pertaining to a protein structural class.

The data analysis used in this study is an unsupervised method; therefore, the resulting components do not necessarily reveal the features directly linked to the classification but represent the sources of variation and the representative properties of the raw data. Even with the small data set presented here, relevant changes in secondary structures of proteins correlate with other, more traditional, label-free methods such as DLS, CD-spectroscopy, and tryptophan fluorescence. As such, identifying relevant changes is possible, but it will require a larger dataset to create a model predicting relevant structural changes based on changes in Raman spectra. Additionally, a higher resolution Raman dataset using more advanced detectors would be beneficial in reducing the noise-to-signal ratio and to improve the PCA analysis.

## 5. Conclusions

In combination with K-means clustering, PCA sheds further light on changes in the structural elements of the Raman spectroscopy spectra. Principal component analysis can be considered a noise filtering method. The relevant differences are captured in the first components, while the higher components contain noise only. The spectra can be reconstructed using only the first p components. The current study demonstrates the capabilities of time-gated Raman spectroscopy in characterizing structural changes of proteins under different experimental conditions without offline sampling and the addition of protein labels. Time-gated Raman spectroscopy gives valuable insights into secondary protein structures that correlate with observations with tryptophan fluorescence spectroscopy, dynamic light scattering, and circular dichroism spectroscopy. Additional variations to perturb proteins could be added as additional parameters to identify additional descriptors of protein unfolding. Raman signals can then be translated into high-level structural information of interest to derive statistical models from being used to predict the relative folding states of unknown samples compared to fully folded proteins. We intend to create additional data sets for building such models in the near future.

## Figures and Tables

**Figure 1 pharmaceutics-14-01639-f001:**
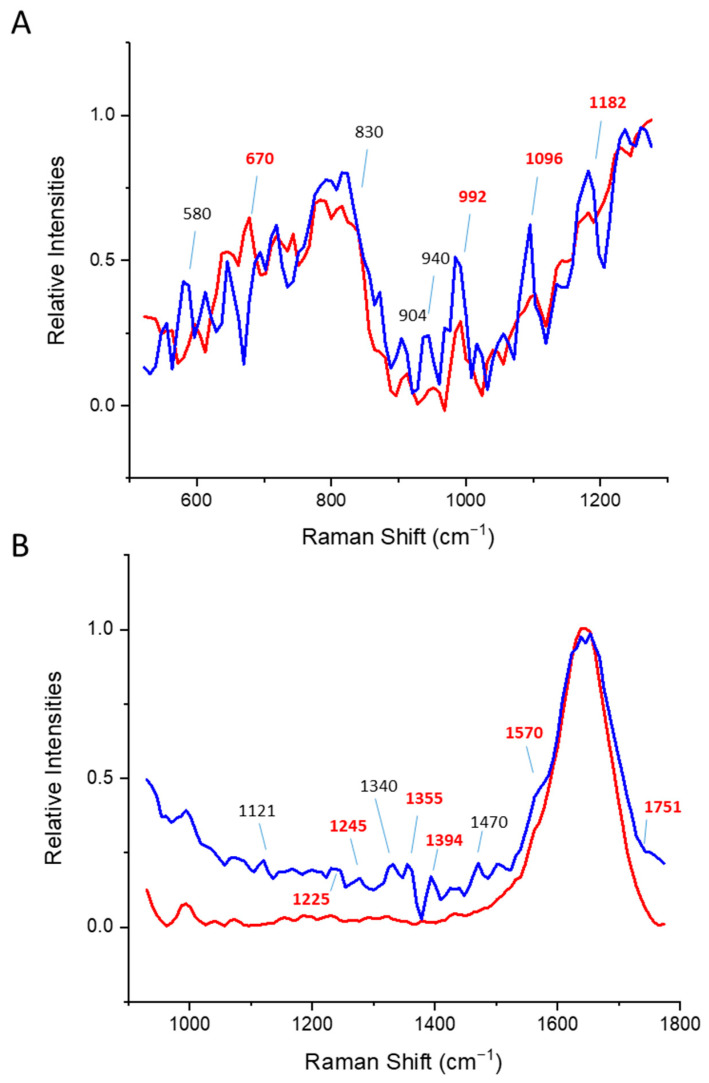
Averaged (N = 11) and normalized time-gated spectra of pepsin at 25 °C (blue) and 65 °C (red). (**A**,**B**) correspond to the individual, non-processed, and non-normalized Time-gated spectra as presented in the left and right panes in Appendix A. (i.e., raw data). Raman values of significance, as presented in Appendix A, are depicted in bold and red.

**Figure 3 pharmaceutics-14-01639-f003:**
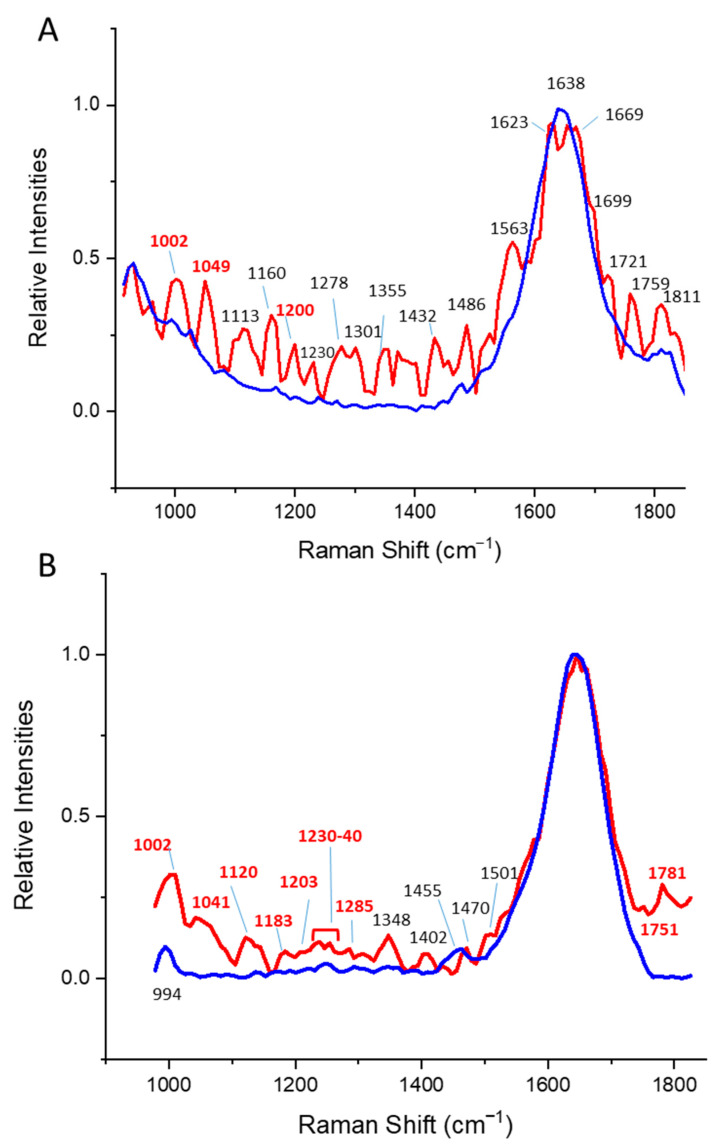
Averaged (N = 11) and normalized Time-gated spectra of (**A**) ovalbumin at 25 °C (blue) and 50 °C (red), and (**B**) *Lm*TIM_E65Q_ at 25 °C (blue) and 86 °C (red) correspond to the individual, non-processed, and non-normalized time-gated spectra as presented in Appendix A (i.e., raw data). Raman values of significance, as presented in Appendix A, are depicted in bold and red.

**Table 1 pharmaceutics-14-01639-t001:** Overview of the proteins used in this study.

Protein (α, β, α/β)	DLS	Tryptophan Fluorescence	CD	Time-Gated ^a^
BSA (α)	yes	yes	yes	yes
CNTF (α)	no ^b^	yes	yes	no ^c^
F_ab_ (β)	yes	yes	yes	no
F_(ab′)2_ (β)	yes	no	no	no
IgG_glycosylated_ (β)	yes	yes	yes	yes
IgG_non-glycosylated_ (β)	yes	no	yes	yes ^d^
Pepsin A (EC 3.4.23.1) (α/β)	yes	no	yes	yes
Ovalbumin (α/β)	yes	yes	yes	yes
*Sc*TIM (EC 5.3.1.1) (α/β)	yes	yes	yes	no
*Lm*TIM_E65Q_ (EC 5.3.1.1) (α/β)	no	yes	yes	yes

^a^ Samples as prepared for tryptophan fluorescence; ^b^ the DLS evaluation of *h*CNTF was recently published in Itkonen et al., 2020 [27]; ^c^ the time-gated evaluation of *h*CNTF protein aggregates was recently published in Kögler et al., 2020 [17]; ^d^ Sample was heated with a custom-built heat unit (Appendix A).

**Table 2 pharmaceutics-14-01639-t002:** Molecular descriptors per analytical technique.

Technique	Parameter 1	Parameter 2	Parameter 3
DLS	Z-average	Hydrodynamic diameter	Polydispersity index
Tryptophan fluorescence	Fluorescence intensity (internal) quenching	Red/blue shift	Tryptophan oxidation (peak at 515 nm)
CD	Melting temperature (°C)	Van’t Hoff enthalpy (kJ/mol)	-
Time-gated Raman spectroscopy	Raman spectra similarity clustering according to temperature	Relevant time-gated Raman peaks ^a^	-

^a^ According to PCA analysis.

## Data Availability

Not applicable.

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
