# Peer review of "Analysis of Biologics Molecular Descriptors towards Predictive Modelling for Protein Drug Development Using Time-Gated Raman Spectroscopy"

_pharmaceutics, 2022, doi:10.3390/pharmaceutics14081639_

Round 1

Reviewer 1 Report

Thanks for the interesting and detailed analysis through Raman spectroscopy 
Something which confused me about the measurement 
1.    There is no detail of the sample preparation for this Raman measurement, whether data from the protein is recorded in the solution of a solid phase (it only says ambient temperature and humidity)
2.    If in solution mode which solvent was used or if in the solid phase, how does it mimic the actual biological environment for the appropriate unfolding of this protein 
3.    I think there is too much detail available in the main manuscript which somehow distracts the reader from the core message of the article. For example, table 3 doesn’t have any meaning in the main text which can go to supporting information.

Author Response

Reviewer 1:

Thanks for the interesting and detailed analysis through Raman spectroscopy 
Something which confused me about the measurement 

  1.    There is no detail of the sample preparation for this Raman measurement, whether data from the protein is recorded in the solution of a solid phase (it only says ambient temperature and humidity).

We appreciate this comment to improve the manuscript. All Raman spectra were recorded with protein in solution (no solid phase), though most of this information was mentioned in the table, we agree it was not clear. We have added the sample preparation to the method section in a more comprehensive way.

  1.    If in solution mode which solvent was used or if in the solid phase, how does it mimic the actual biological environment for the appropriate unfolding of this protein 

Since the aim of the study is to evaluate which molecular descriptors are representative for protein unfolding when comparing non-normalized Raman spectra, proteins were diluted in 1 x phosphate saline buffer at neutral pH. These molecular descriptors of unfolding (e.g. changes in secondary structures) would then be a useful tool when evaluating proteins in-vivo, during secretion (e.g. by a fungal host), during down-stream processing and during storage in buffered solution. We did not aim in this study to represent in vivo conditions, but develop a monitoring tool for pharmaceutical protein development, production and storage.  

  1.    I think there is too much detail available in the main manuscript which somehow distracts the reader from the core message of the article. For example, table 3 doesn’t have any meaning in the main text which can go to supporting information.

We agree on moving tables such as table 3 – 8 from the main text and placing these in the supplementary data to shorten the manuscript and avoid distractions. In addition, we have re-written the introduction and discussion to focus more on the core message. All changes in the text are shown in dark red, the changed manuscript is attached as a PDF. 

Reviewer 2 Report

The authors want to monitor the conformation change of proteins directly using the time-gated Raman method. This is a big goal for chemists and biologists. However, Raman signals of proteins are so weak that people could hardly obtain the structure information of proteins. Thus, the authors introduce the analytic methods to figure out the structure change of proteins. However, the quality of their Raman spectra is poor; several peaks are so small to be significantly differentiated from the background noises—for example, the signal at 1225 cm-1 in Fig. 1B. In addition, many typos can be found in this manuscript. For example, there is no Fig 21 corresponding to what the authors describe (line 419). And, most of the wavenumber was not present in the correct format. Although this study seems to pave the way for in-line and label-free methods to study the protein change in cells, the manuscript was not easy to understand. Maybe the manuscript covers too many things and diverse their target. I would suggest that they need focus on fewer proteins and interpretate their data in a definite way.

Author Response

Reviewer 2:

The authors want to monitor the conformation change of proteins directly using the time-gated Raman method. This is a big goal for chemists and biologists. However, Raman signals of proteins are so weak that people could hardly obtain the structure information of proteins. Thus, the authors introduce the analytic methods to figure out the structure change of proteins. However, the quality of their Raman spectra is poor; several peaks are so small to be significantly differentiated from the background noises—for example, the signal at 1225 cm-1 in Fig. 1B. In addition, many typos can be found in this manuscript. For example, there is no Fig 21 corresponding to what the authors describe (line 419). And, most of the wavenumber was not present in the correct format. Although this study seems to pave the way for in-line and label-free methods to study the protein change in cells, the manuscript was not easy to understand. Maybe the manuscript covers too many things and diverse their target. I would suggest that they need focus on fewer proteins and interpretate their data in a definite way.

We agree with the reviewer that the quality of the Raman data could be improved by using the newer TimegatedTM – Raman spectroscope (Timegated Ltd (Oulu); 2nd generation). However, we have only the first generation device available with the lower resolution detector. The Raman data, as presented in the supplementary data are the raw, time-gated Raman spectra without smoothing. The statistical methods applied do highlight significant changes in the peaks, even the smaller ones. In the past we would analyse the changes in the peaks visually and this miss these changes. As such, we agree that the small peaks seem insignificant, but in these experiments significant changes due to thermal unfolding were observed.

The signal-to-noise ratio (S/N) was obtained by looking at the raw data in more detail. For example, when using the IgG data and focusing on the areas without major peaks, where the regions from 640 - 740 cm-1 and 1720 - 1820 cm-1 are a good example to observe that the noise level is around 25 counts. This “noise” is approximately the detector noise, and not from the sample and it is fairly low compared to the other peaks.

See plots below:

We did not measure any proteins inside cells (we already showed that Raman spectroscopy can be used to measure protein expression in E. coli cells in a previous study (Kögler , M , Itkonen , J , Viitala , T & Casteleijn , M G 2020 , ' Assessment of recombinant protein production in E. coli with Time-Gated Surface Enhanced Raman Spectroscopy (TG-SERS) ' , Scientific Reports , vol. 10 , no. 1 , 2472 . https://doi.org/10.1038/s41598-020-59091-3). In the current study presented here, we only examined the thermal unfolding of proteins and the changes in the time-gated Raman spectra in a buffered solution (i.e. in vitro).

We thank the reviewer for pointing out mistakes in the manuscript and had a second native English speaker evaluate the manuscript on spelling errors. We also removed tables 3-8 and moved these to the supplementary data file. We refer to figures in the supplementary data with the addition of the letter “S” to the figure number, thus while it is true there is not figure 21, we are referring to figure S21 which can be found in the Supplementary data file.

Overall, the research idea was to evaluate several proteins in order to find common molecular descriptors within time-gated Raman spectra for thermal unfolding of proteins and compare the changes in Raman data with established spectroscopic methods often used to evaluate changes in the secondary structure of proteins and/or the formation of aggregates. Though, for each protein a deeper analysis using NMR unfolding studies and additional unfolding studies using different unfolding methods would be very insightful for each specific protein, this was not aim of these experiments.  

ll changes in the text are shown in dark red in the revised manuscript attached as PDF.

Reviewer 3 Report

Title:   Analysis of biologics molecular descriptors towards predictive modelling for protein drug development using time-gated Raman spectroscopy

The paper by the Castelijn lab presents a methodology based on time-gated Raman spectroscopy for the detection of protein unfolding and aggregation for predictive modeling of protein drug development. The authors aim to evaluate the unfolding of proteins characterized by main differences in secondary structures (i.e., α-helix, β-sheet and, α/β) in comparison with state-of-the-art techniques, including tryptophan fluorescence, circular dichroism, and dynamic light scattering that monitor the aggregation, unfolding, and melting temperature. Chemometric methods including PCA and k-means clustering were employed to identify the molecular spectral variability and the differentiation of protein’s structural components.

While the scientific idea is very interesting, the interpretation of the presented data needs to be improved. Fundamental basics in TG-Raman spectroscopy are unclear, and the presented (chemometric) data after correction is overinterpreted. The manuscript is poorly written; its methods section does not describe how TG-Raman spectroscopy was pursued on protein samples. Due to these points, I do not recommend the publication of the current manuscript.

Major issues:

  1. A detailed experimental section on TG-Raman spectroscopy is missing, in which the protein sample preparation procedure for Raman, measurement, evaluation, and statistics are laid out. Please re-write this section:
  • Setup: Specify how the TG-Raman experiments were technically carried out. Which light source (characteristics, i.e., wavelength, pulse length, spectral width) has been used? Which grating was chosen, i.e., what is the spectral resolution of the system? What is the minimal S/N ratio of the presented data for the given acquisition settings at different temperatures?
  • Protein spectra: Please describe the protein sample preparation, measurement, evaluation, and statistics of the pure protein Raman spectra. What was the concentration of each sample, the total amount loaded to the system? How many repeats per experimental condition were carried out? How were protein samples studied? Consecutively/parallel? What are the underlying statistics of the presented data? How about their statistical significance? What is the minimal S/N ratio for data that was still taken into account for the analysis? What is the error of the assigned Raman resonances in amplitude and central frequency?
  • Section 2.5: With respect to data pre-processing, the authors are encouraged to check the background-subtraction procedure. As seen e.g. in SI Figure S5, the Raman amplitude is artificially rising from 1100 cm-1 on in Raman scans between 400-1300 cm-1, while their baseline is flat for scans between 900-1800 cm-1. Similarly, there is a baseline artifact in 900-1800 cm-1 scans below 1100 cm-1.

  1. The presented (chemometric) data is currently overinterpreted.
    • Following the SI data, the spectral resolution and signal-to-noise ratio of the data are already low at room temperature. Raman spectra at higher temperatures suffer from even lower quality (cf. for example Figure 2A). Due to the strong noise, smoothing with a higher-order Savgol Filter (5/7) will lead to artifacts (cf. the Raman resonance at 1631 cm-1 in Figure 2A at room temperature and 65 oC). The assigned Raman peaks are overinterpreted and not visible within the presented heat-dependent data in the SI.
    • A short comment on PCA: While k-mean clustering sorts ‘unaltered’ spectra according to their similarity, PCA loadings do not resemble actual Raman spectra. Observed peak changes do not report on actual molecular resonance or changes their off, and should not be interpreted (in particular not with these strong smoothing artifacts. Please reconsider the results summarized in the tables per each protein structure section.
    • The authors present results from k-mean clustering and PCA for 3 different secondary protein structures. Please summarize your findings at the end of each section in order to identify molecular descriptors of protein unfolding for the different categories. Are there similarities? What do we learn from TG-Raman? What are the strength and weaknesses?

  1. Due to the missing experimental section and imprecision in the discussion section, it is unclear which kind of Raman spectroscopy the authors actually carried out.
  • Page 10 and 16: The authors describe in the abstract and main part of the paper to have used time-gated Raman spectroscopy, however, on pages 10 and 16, they refer to it as TG-SERS. SERS like Surface-enhanced Raman scattering or pure Spontaneous Raman scattering in solution? If it is SERS - it is not appropriately described in the methods section. Which SERS experiments have been carried out? What kind of substrate has been used? Seeing the presented spectra, however, it seems the authors mixed acronyms.
  • Page 10 / l. 331-332: Which structural feature in IgG leads to SERS in both proteins? What do the author mean? An enhancement/quenching of the Raman signature? Enhancement due to Resonance Raman due to electronic transitions within the studied molecule?

  1. The Supplementary information is poorly presented.
  • In general: please enlarge the labeling size of the presented spectra in all plots for better readability. Moreover, please use high-resolution dpi figures for better readability of the plots.
  • Figure S3 and S8B: please comment or mark the potential outliers in Z-Average plots of the DLS data of BSA and pepsin.
  • In tryptophan assays (S4, S18), linear changes in temperature are encoded in rainbow coloring, however, the trends are very hard to see. Could the authors maybe label the traces with colored temperatures, or use arrows to indicate the directional trends and changes in the spectral plots to guide the reader?
  • Please define the y-axis used in the t-dependent TG-Raman spectra. How are the spectra shifted against each other?

Minor issues to improve the manuscript:

  • Small comments
  • 6 / l.246: In times of open access and scientific transparency, it is good practice to show all data.
  • Section 3.1: it is difficult to follow the results found for BSA compared to CNTF when jumping between techniques. I suggest to discussing the experimental findings per protein and highlighting potentially found spectral descriptors for proteins with alpha-helices at the end of the section.
  • Formatting
    • Please refer to Figures in the SI chronologically: e.g. on page 7 line 277/278, there is a jump between Figure S10 to S13.
    • 2 / l.92: Please add a header for the chemicals and reagents section.

  • Spelling:
  • The authors change between English and American English. See for example on the first page: Labelling, modelling vs. anemia, behavior, unfavorable. Please check the manuscript on consistent language use and grammar.
  • Please ensure consistent notation, e.g., protein drug vs. protein-drugs. Raman vs. raman,

  • Typos / unclear notations:
  • 3 / l.116/117. Please correct the sample volume and syringe filter size.
  • 3 / l.111. Please units in the statement: “peak at 515”
  • 6 / l.209: Please specify what the notation var (PC1) $ var (PC2) $ var (PCp) means.
  • 6 / l.242: Please correct the notation of the 310 alpha-helix structure to 3_{10} ..
  • 7 / l.272: Please add details, like item numbers “EC 3.4.23.1” to the materials section and remove it from the main text.
  • 7 / l. 276. Please correct to ” … starts aggregating …”
  • 10 / l. 335: Please correct the spelling of “vSO4” according to Raman notation!
  • 17 / Table 11: Please specify the abbreviations “ND”.
  • Please check the manuscript for misinterpreted Greek symbols, like µ e.g. on 4./l129; p. 4/l.157; p. 4./l.166 …
  • Please ensure correct writing of wavenumber units: cm^{-1}!
  • SI: Figure caption S1: Please correct type: “Blue arrows depitc the water flow, …”

Author Response

reviewer 3:

The paper by the Castelijn lab presents a methodology based on time-gated Raman spectroscopy for the detection of protein unfolding and aggregation for predictive modeling of protein drug development. The authors aim to evaluate the unfolding of proteins characterized by main differences in secondary structures (i.e., α-helix, β-sheet and, α/β) in comparison with state-of-the-art techniques, including tryptophan fluorescence, circular dichroism, and dynamic light scattering that monitor the aggregation, unfolding, and melting temperature. Chemometric methods including PCA and k-means clustering were employed to identify the molecular spectral variability and the differentiation of protein’s structural components.

While the scientific idea is very interesting, the interpretation of the presented data needs to be improved. Fundamental basics in TG-Raman spectroscopy are unclear, and the presented (chemometric) data after correction is overinterpreted. The manuscript is poorly written; its methods section does not describe how TG-Raman spectroscopy was pursued on protein samples. Due to these points, I do not recommend the publication of the current manuscript.

 Major issues:

  1. A detailed experimental section on TG-Raman spectroscopy is missing, in which the protein sample preparation procedure for Raman, measurement, evaluation, and statistics are laid out. Please re-write this section:
  • Setup: Specify how the TG-Raman experiments were technically carried out. Which light source (characteristics, i.e., wavelength, pulse length, spectral width) has been used? Which grating was chosen, i.e., what is the spectral resolution of the system? What is the minimal S/N ratio of the presented data for the given acquisition settings at different temperatures?

The experimental set-up was partially the same as previously described in Kögler, M , Itkonen , J , Viitala , T & Casteleijn , M G 2020 , ' Assessment of recombinant protein production in E. coli with Time-Gated Surface Enhanced Raman Spectroscopy (TG-SERS) ' , Scientific Reports , vol. 10 , no. 1 , 2472 . https://doi.org/10.1038/s41598-020-59091-3.), without the use of a microscope, however all other parameters are kept the same. The details of the device are reported before in Tiina Lipiäinen’s 2018 time gated-Raman paper where they also used the “old M1 detector“. We have removed reference 31 and included this reference in the method section:

https://pubs.acs.org/doi/abs/10.1021/acs.analchem.8b00298. In addition, reference 30: Kostamovaara, J.; Tenhunen, J.; Kögler, M.; Nissinen, I.; Nissinen, J.; Keränen, P. Fluorescence suppression in Raman spec-troscopy using a time-gated CMOS SPAD. Optics Express 2013, 21, 31632-31632, oi:10.1364/OE.21.031632 gives additional insights.

The signal-to-noise ratio (S/N) was obtained by looking at the raw data in more detail. For example, when using the IgG data and focusing on the areas without major peaks, where the regions from 640 - 740 cm-1 and 1720 - 1820 cm-1 are a good example to observe that the noise level is around 25 counts. This “noise” is approximately the detector noise, and not from the sample and it is fairly low compared to the other peaks.

See plots below:

We evaluate the S/N also at higher temperatures in a similar way and see the same values. Heating of the protein sample does not affect the S/N. The detector noise is only affected by the dark count or the electronic noise from the detector. The CMOS SPAD detectors used in the Time-gate system’s have very little dark count or detector noise and they do not need cooling like the common CCD based systems.

  • Protein spectra: Please describe the protein sample preparation, measurement, evaluation, and statistics of the pure protein Raman spectra. What was the concentration of each sample, the total amount loaded to the system? How many repeats per experimental condition were carried out? How were protein samples studied? Consecutively/parallel? What are the underlying statistics of the presented data? How about their statistical significance? What is the minimal S/N ratio for data that was still taken into account for the analysis? What is the error of the assigned Raman resonances in amplitude and central frequency?

All Raman spectra were recorded with protein in solution, though this information was mentioned in the table, we agree it was not clear. We have added the sample preparation to the method section. Since we aim to mimic experimental conditions where sampling is not possible, we have observed changes in the raw data of the spectra and evaluate these changes with unsupervised k-means clustering and PCA methods to identify significant changes. Though we agree this method is cruder than building a data-set from several protein preparations, we argue that with our approach we do see relevant changes due to thermal protein unfolding. Each spectrum is an average of 11 measurements taken. The software provided by Timegated Ltd. subtracts the background noise during the post-processing. The S/N in the timegated postprocessing software was the same as described earlier by Lipiäimen et al (2018): (https://pubs.acs.org/doi/abs/10.1021/acs.analchem.8b00298. We did not perform a baseline correction thereafter, nor did we apply smoothing.

  • Section 2.5: With respect to data pre-processing, the authors are encouraged to check the background-subtraction procedure. As seen e.g. in SI Figure S5, the Raman amplitude is artificially rising from 1100 cm-1 on in Raman scans between 400-1300 cm-1, while their baseline is flat for scans between 900-1800 cm-1. Similarly, there is a baseline artifact in 900-1800 cm-1 scans below 1100 cm-1.

The effect reviewer 3 is referring to seems only be obvious when intensities are low. These effects are due to the detector and the baseline is not corrected with the Timegated software, nor afterwards (e.g. in Origin or MatLab). We intentionally do not correct for the baseline for this analysis, since we are aiming for a method that can be utilized in the future in an automated way by only evaluating the raw data. However, if changes in this region were significant in both the low and the higher wavenumber scans, the PCA analysis will isolate these changes as significant. The difference in intensity between the low and the higher wavenumber scans (11 repeats / Raman spectrum) of the same sample is due to the higher intensity from the Amide I peak during the second scan. In this first generation time-gated Raman device from TimeGated Ltd, only a smaller section of a spectrum can be measured when opting for higher resolution (steps between lowest and highest wavenumber measured). In the non-glycosylated samples we opted for larger steps to cover most the same spectrum, but here we have less repetitions per scan and the steps were a bit larger. Thus, the effect seen is due to the differences in the intensities of these non-normalized data sets. We have added this issue in the discussion.  

  1. The presented (chemometric) data is currently overinterpreted.
    • Following the SI data, the spectral resolution and signal-to-noise ratio of the data are already low at room temperature. Raman spectra at higher temperatures suffer from even lower quality (cf. for example Figure 2A). Due to the strong noise, smoothing with a higher-order Savgol Filter (5/7) will lead to artifacts (cf. the Raman resonance at 1631 cm-1 in Figure 2A at room temperature and 65 oC). The assigned Raman peaks are overinterpreted and not visible within the presented heat-dependent data in the SI.

The R hyperSpec package provides several functions for smoothing interpolation, background correction, offset correction, baseline correction, but none of these gave useful result with our dataset. Therefore, we didn’t used them before the PCA, and we choose to present PCA result anyway. Savitsky-Golay filter (5th order, 7 points) was the only option that didn’t affect too much the date for further modelling.  During the review of the comments, we prepare the data for PCA without using the Savitsky-Golay filter (5th order, 7 points) and this didn’t change the result obtained for the most of the first components. So, the sentence is deleted from the text. It seems the available smoothing method doesn’t help with our dataset. Principal component analysis can be considered as a noise filtering method. The relevant differences are captured in the first components while the higher components contain noise only. The spectra can be reconstructed using only the first p components.

Regarding the signals due to heating of the samples, we observe a sudden loss in intensities, especially of the amide I peak region, thus when these spectra are normalized and overlayed larger chances in the heated sample can be expected. However, all data that is presented in the supplementary figures was used for the k-means clustering and PCA analysis.

    • A short comment on PCA: While k-mean clustering sorts ‘unaltered’ spectra according to their similarity, PCA loadings do not resemble actual Raman spectra. Observed peak changes do not report on actual molecular resonance or changes their off, and should not be interpreted (in particular not with these strong smoothing artifacts). Please reconsider the results summarized in the tables per each protein structure section.

The data used for PCA loading was not smoothed (see comment above). We used the raw data, as is presented in the supplementary data file. In addition, PCA / PLS over the whole spectral range is common procedure in spectral analysis (NIR and Raman), as can be seen in this example: https://link.springer.com/article/10.1007/s11356-018-2180-2

    • The authors present results from k-mean clustering and PCA for 3 different secondary protein structures. Please summarize your findings at the end of each section in order to identify molecular descriptors of protein unfolding for the different categories. Are there similarities? What do we learn from TG-Raman? What are the strength and weaknesses?

We appreciate this comment and we have added sections in the text. All changes in the text are shown in dark red.

  1. Due to the missing experimental section and imprecision in the discussion section, it is unclear which kind of Raman spectroscopy the authors actually carried out.
  • Page 10 and 16: The authors describe in the abstract and main part of the paper to have used time-gated Raman spectroscopy, however, on pages 10 and 16, they refer to it as TG-SERS. SERS like Surface-enhanced Raman scattering or pure Spontaneous Raman scattering in solution? If it is SERS - it is not appropriately described in the methods section. Which SERS experiments have been carried out? What kind of substrate has been used? Seeing the presented spectra, however, it seems the authors mixed acronyms.

We thank the reviewer to point this error out to us and we have removed and surface enhanced Raman Spectroscopy references in the text. We also included the correct reference to what kind of Raman spectroscopy was used (by Lipiäimen et al (2018): (https://pubs.acs.org/doi/abs/10.1021/acs.analchem.8b00298)

  • Page 10 / l. 331-332: Which structural feature in IgG leads to SERS in both proteins? What do the author mean? An enhancement/quenching of the Raman signature? Enhancement due to Resonance Raman due to electronic transitions within the studied molecule?

We thank the reviewer to point this error out to us and we have removed this conclusion.

 The Supplementary information is poorly presented.

  • In general: please enlarge the labeling size of the presented spectra in all plots for better readability. Moreover, please use high-resolution dpi figures for better readability of the plots.

We agree with the reviewer and have made these improvements.

  • Figure S3 and S8B: please comment or mark the potential outliers in Z-Average plots of the DLS data of BSA and pepsin.

We have add comments to the figures.

  • In tryptophan assays (S4, S18), linear changes in temperature are encoded in rainbow coloring, however, the trends are very hard to see. Could the authors maybe label the traces with colored temperatures, or use arrows to indicate the directional trends and changes in the spectral plots to guide the reader?

This is a good suggestion and we have added the labels in the figures.

  • Please define the y-axis used in the t-dependent TG-Raman spectra. How are the spectra shifted against each other?

The spectra are not shifted against each other. The different spectra at different temperatures are stacked above each other for clarity and to compare the spectra in one representation. We have added this clarification to the figures.   

Minor issues to improve the manuscript:

  • Small comments
  • 6 / l.246: In times of open access and scientific transparency, it is good practice to show all data.
  • Section 3.1: it is difficult to follow the results found for BSA compared to CNTF when jumping between techniques. I suggest to discussing the experimental findings per protein and highlighting potentially found spectral descriptors for proteins with alpha-helices at the end of the section.

Thank you for the suggestion. We have changed the manuscript accordingly.

  • Formatting
    • Please refer to Figures in the SI chronologically: e.g. on page 7 line 277/278, there is a jump between Figure S10 to S13.

This has been addressed.

    • 2 / l.92: Please add a header for the chemicals and reagents section.

We have added the header.

 Spelling:

  • The authors change between English and American English. See for example on the first page: Labelling, modelling vs. anemia, behavior, unfavorable. Please check the manuscript on consistent language use and grammar.

We thank the reviewer for pointing out mistakes in the manuscript and had a second native English speaker evaluate the manuscript on spelling errors.

  • Please ensure consistent notation, e.g., protein drug vs. protein-drugs. Raman vs. raman,

Thank you for pointing this out. We have made the changes.

  • Typos / unclear notations:

Thank you for pointing this out. We have made the changes regarding the typos and unclear notations listed below:

  • 3 / l.116/117. Please correct the sample volume and syringe filter size.
  • 3 / l.111. Please units in the statement: “peak at 515”
  • 6 / l.209: Please specify what the notation var (PC1) $ var (PC2) $ var (PCp) means.
  • 6 / l.242: Please correct the notation of the 310 alpha-helix structure to 3_{10} ..
  • 7 / l.272: Please add details, like item numbers “EC 3.4.23.1” to the materials section and remove it from the main text.
  • 7 / l. 276. Please correct to ” … starts aggregating …”
  • 10 / l. 335: Please correct the spelling of “vSO4” according to Raman notation!
  • 17 / Table 11: Please specify the abbreviations “ND”.
  • Please check the manuscript for misinterpreted Greek symbols, like µ e.g. on 4./l129; p. 4/l.157; p. 4./l.166 …
  • Please ensure correct writing of wavenumber units: cm^{-1}!
  • SI: Figure caption S1: Please correct type: “Blue arrows depict the water flow, …”

Round 2

Reviewer 3 Report

I am happy to see that the manuscript has improved a lot. The methods / research design are well described. All raised concerns and suggestions have been addressed and incorporated. Also, the language did improve (despite some typos). I, therefore, recommend the submitted manuscript for publication in Pharmaceutics.

I encourage the authors to proofread against typos and spellings, e.g. on:

* page 4, Line 143: "... data between at 22C and 90 and 280 nm..."

* page 11, Line 384: "summering" = summarising 

* page 11, Line 398: "tryptophanE" ... check for consistency in the MS 

same on p 15/L551